# Sonodynamic Therapy for HER2+ Breast Cancer with Iodinated Heptamethine Cyanine–Trastuzumab Conjugate

**DOI:** 10.3390/ijms251810137

**Published:** 2024-09-21

**Authors:** Dmytro Kobzev, Olga Semenova, Sarit Aviel-Ronen, Olesia Kulyk, Raanan Carmieli, Tajib Mirzabekov, Gary Gellerman, Leonid Patsenker

**Affiliations:** 1Department of Chemical Sciences, Faculty of Natural Sciences, Ariel University, Ariel 40700, Israel; dmytrok@ariel.ac.il (D.K.); olgasem@ariel.ac.il (O.S.); olesiak@ariel.ac.il (O.K.); garyg@ariel.ac.il (G.G.); 2Adelson School of Medicine, Ariel University, Ariel 40700, Israel; saritav@ariel.ac.il; 3Sheba Medical Center, Tel-Hashomer, Ramat Gan 52621, Israel; 4Department of Chemical Research Support, Weizmann Institute of Science, Rehovot 76100, Israel; raanan.carmieli@weizmann.ac.il; 5Biomirex, Inc., 27 Strathmore Road, Natick, MA 01760, USA; tmirzabekov@biomirex.com

**Keywords:** targeted sonodynamic therapy, sonodynamic therapy, photodynamic therapy, HER2-positive breast cancer, trastuzumab, antibody–sonosensitizer conjugate, iodinated cyanine, reactive oxygen species

## Abstract

The first example of sonodynamic therapy (SDT) with a cyanine dye–antibody conjugate is reported. The aim of this study was to evaluate the sonodynamic efficacy of a trastuzumab-guided diiodinated heptamethine cyanine-based sensitizer, **2ICy7–Ab,** versus its non-iodinated counterpart, **Cy7–Ab**, in a human epidermal growth factor receptor 2-positive (HER2+) xenograft model. In addition, the combined sonodynamic and photodynamic (PDT) effects were investigated. A single intravenous injection of **2ICy7–Ab** followed by sonication or combined sonication and photoirradiation in mice resulted in complete tumor growth suppression compared with the nontreated control and showed no detectable toxicity to off-target tissues. In contrast, **Cy7–Ab** provided only a moderate therapeutic effect (~1.4–1.6-fold suppression). SDT with **2ICy7–Ab** resulted in a 3.5-fold reduction in tumor volume within 45 days and exhibited 13-fold greater tumor suppression than PDT alone. In addition, **2ICy7–Ab** showed more durable sonostability than photostability. The sonotoxicity of the iodinated versus noniodinated counterparts is attributed to the increased generation of hydroxyl radicals, superoxide, and singlet oxygen. We observed no significant contribution of PDT to the efficacy of the combined SDT and PDT, indicating that SDT with **2ICy7–Ab** is superior to PDT alone. These new findings set the stage for the application of cyanine–antibody conjugates for fluorescently monitored targeted sonodynamic treatment of cancer.

## 1. Introduction

Sonodynamic (**SDT**) and photodynamic (**PDT**) therapies are established non-invasive therapeutic modalities [1] for treating cancer [2,3] and various other diseases [4,5,6,7]. Both methods utilize *sensitizers*, which are, in general, organic dye molecules or nanoparticles [8,9,10] that act as sonosensitizers or photosensitizers, depending on the nature of the excitation energy, ultrasound (US), or light irradiation, respectively. Sensitizers are typically administered intravenously (IV) [11,12] to reach malignant cells and, upon US or light irradiation, produce cytotoxic species, such as reactive oxygen species (ROS) [8,13], that kill those cells. The mechanism underlying SDT and a comparison with PDT have been described in detail elsewhere [13]. Despite the generally similar principles of action (excitation of sensitizer followed by the generation of cytotoxic species) [13], ultrasound penetrates much deeper into the body (20–30 cm) [13,14,15,16] than NIR light (~2 cm) [17,18], making SDT more effective than PDT, particularly for primary tumors and metastatic cancers that are located deep within the body [11,12,19]. However, SDT is considerably less developed than PDT, and only a few photosensitizers, specifically porphyrins [10,20,21,22], phthalocyanines [23,24,25], xanthenes [4,26,27,28,29], and only two *cyanines* (indocyanine green, **ICG** [30,31,32], and **IR780** [33,34]), have been successfully employed for treatment with ultrasound excitation [35,36,37].

Importantly, it is not evident that a dye molecule acting as a photosensitizer can be used effectively as a sonosensitizer. This can be attributed to the lack of dye stability under sonication or the insufficient quantum yield of reactive species production upon sonication compared with light [38].

A major limitation for the therapeutic utility of cyanine-based sono- and photosensitizers is the low yield of generated reactive species and, as a result, insufficient treatment efficacy. The incorporation of *heavy atoms*, such as iodine, bromine, sulfur, and selenium, was recently proven to enhance the PDT efficiency of cyanines due to increased spin-orbit coupling and, therefore, intersystem crossing from the singlet to the triplet state, which results in elevated rates of cytotoxic species generation [39,40]. Thus, the introduction of iodine atoms causes a noticeable increase in the ability of cyanine dyes to photokill cancer cells [41,42,43], as well as Gram-positive and Gram-negative pathogenic bacteria [44,45,46,47].

However, currently available sonosensitizers, including cyanines, suffer from the same limitation as photosensitizers: their targeting specificity to malignant cells is insufficient, leading to side effects, which hinders their prevalent clinical use. This issue can be resolved by equipping the sensitizer with target-specific carriers, such as antibodies [32,48,49,50,51,52]. Despite pronounced progress in the development and application of cyanine dyes in PDT that has been achieved in recent years [53], the potential of cyanines for use in SDT has still been poorly explored, whereas cyanine–antibody conjugates for SDT applications have yet to be investigated.

Recently, we developed an antibody-guided diiodinated heptamethine cyanine-based **2ICy7–Ab** conjugate, where the antibody (Ab) is trastuzumab, and demonstrated the high efficiency of this conjugate in the PDT of human epidermal growth factor receptor 2-positive (HER2+) human breast cancer in a xenograft mouse model [41]. In this work, we explored the ability of this **2ICy7–Ab** conjugate for SDT and combined PDT + SDT to suppress tumor growth, and we compared the performance of this conjugate with its non-iodinated **Cy7–Ab** counterpart.



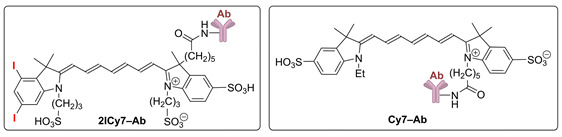



## 2. Results and Discussion

### 2.1. Photostability and Sonostability

Organic dyes can degrade upon light and ultrasound irradiation [54,55,56], which can reduce their photo- and sonodynamic efficacy. Therefore, we evaluated the photo- and sonostability of the iodinated vs. non-iodinated dyes and their Ab conjugates. To this end, we irradiated solutions of these compounds in phosphate buffer saline (PBS) (pH 7.4), measured the time-dependent absorption and emission spectra of the solutions vs. the applied light (Appendix A) and ultrasound (US, Appendix A) doses, and plotted the corresponding decay curves (Figure 1). Then, the stabilities of the dyes and conjugates toward the light and US irradiation were quantified through their degradation half-lives (τ_1/2,Light_ and τ_1/2,US_; Table 1). On the basis of τ_1/2,Light_ and τ_1/2,US_, we calculated the irradiation light (*D*_1/2,Light_) and US (*D*_1/2,US_) doses required for 50% degradation of the dyes and Ab conjugates.

In our further experiments with mice (see below), we applied a US power density of 0.7 W/cm^2^ (frequency f = 1 MHz), which is safe for animals [57], and a light irradiance of 8 mW/cm^2^ (730 nm LED) [41]. These conditions were used to investigate the sono- and photostabilities of the dyes and conjugates. It can be seen from the presented data (Figure 1 and Table 1) that the sonostability of the dyes and conjugates at 0.7 W/cm^2^ is higher than their photostability at 8 mW/cm^2^: a much higher (~10–46 time) US dose (*D*_1/2,US_) is required for their decomposition compared to the light dose (*D*_1/2,Light_). Dye conjugation to an antibody increases photostability and sonostability by approximately 1.2–4.9-fold. Iodination may have different effects: **2ICy7** and **2ICy7–Ab** are less photostable than their non-iodinated analogs; the sonostability of **2ICy7–Ab** is also lower than that of **Cy7–Ab**, whereas **2ICy7** is more stable toward sonication than **Cy7**. As a result, no clear correlation between the sono- and photostability of the investigated compounds was found (Appendix A).

### 2.2. Singlet Oxygen Generation

Recently, we reported on the quantum yields of singlet oxygen (^1^O_2_) generated from **Cy7** and **2ICy7** dyes and their Ab conjugates under light irradiation (Φ_Δ,Light_) [41]. It was found that Φ_Δ,Light_ noticeably (~3-fold) increased upon iodination but decreased 1.7-fold upon conjugation to the Ab (Table 1). The decrease in the Φ_Δ,Light_ of **2ICy7–Ab** compared with that of **Cy7–Ab** was attributed to the aggregation of the iodinated dye on the antibody [41].

In this work, we investigated the efficacy of US-induced ^1^O_2_ generation by these sensitizers. Determination of the absolute quantum yields upon US excitation (Φ_Δ,US_) is problematic [58]. We were also unable to find in the literature any dyes with known Φ_Δ,US_ values to be used as a reference for our measurements. Therefore, the sonodynamic efficacies were estimated in this work through the relative quantum yields (φ_Δ,US_) in comparison with those of **Cy7**, where φ_Δ,US_ of **Cy7** was taken as 1 a.u. The φ_Δ,US_ measurements were carried out in PBS using the same singlet oxygen sensor green (SOSG)-based method as we applied for the Φ_Δ,Light_ measurements [41], with sonication rather than light excitation being used.

The obtained quantum yields of ^1^O_2_ generation (φ_Δ,US_) are shown in Table 1. Upon iodination, φ_Δ,US_ increased 1.6-fold for the free dyes and 2.1-fold for the conjugates. Consequently, the magnitudes of these intensifications are less prominent than those upon light excitation (Φ_Δ,Light_). Additionally, while the Φ_Δ,Light_ of the free dyes decreased 1.7-fold after conjugation, the φ_Δ,US_ increased 3.2- and 4.2-fold for **Cy7**/**Cy7–Ab** and **2ICy7**/**2ICy7–Ab**, respectively. As a result, while the Φ_Δ,Light_ of **2ICy7–Ab** was only 1.8-fold greater than that of **Cy7**, its φ_Δ,US_ was 6.7 times higher. Although the singlet oxygen generation quantum yield does not dramatically increase upon iodination (only 2.1-fold for the conjugate), it contributes to a large increase in SDT efficacy in the mouse model, as demonstrated below.

### 2.3. Detection and Quantification of Radicals

The ability of **2ICy7** vs. **Cy7** to produce free radicals upon sonication and light irradiation and the types of generated reactive species were studied by electron paramagnetic resonance (EPR) spectroscopy (Figure 2, Appendix A). The measurements were performed under 30 min of light irradiation (730 nm, 30 W LED, 8 mW/cm^2^) or after 15 min of sonication (40 kHz, 50 W) of 20 μM aqueous dye solutions in the presence of a 25 mM 5-tert-butoxycarbonyl-5-methyl-1-pyrroline-*N*-oxide (BMPO) spin trap in double distilled water. The EPR spectra of **2ICy7** (black traces) and **Cy7** (red traces) upon light irradiation and sonication are displayed in Figure 2A,B, respectively. The EPR spectrum of the BMPO spin trap is centered at approximately *g* = 2 and exhibits four lines, which is typical of reactions with hydroxyl radicals (^•^OH).

Since the EPR signal intensity is directly related to the dye concentration, Figure 2B shows that, upon sonication, **2ICy7** generates approximately three times more oxygen radicals than **Cy7**. Under light irradiation, however, both dyes can be seen to produce a similar amount of these reactive species (Figure 2A). Using a calibration curve, we calculated the molar concentrations of the generated radicals (*c*_R_) and confirmed that, upon sonication, **2ICy7** indeed produced three times more radicals than **Cy7**, whereas upon light irradiation, this difference was only approximately 10% (Table 2).

To quantify the generated hydroxyl radicals, the measurements were carried out in the presence of 10% dimethyl sulfoxide (DMSO), which acts as an ^•^OH scavenger. Appendix A show a comparison of the EPR spectra recorded with and without DMSO in aqueous solution. It can be seen that the signal intensity decreased in the presence of DMSO but not entirely, indicating that the EPR signal only partially originated from the direct formation of ^•^OH or the spontaneous conversion of BMPO/^•^OOH (superoxide or hydroperoxyl radicals) to BMPO/^•^OH (hydroxyl radical) [59]. The data in Table 2 show that upon sonication, **2ICy7** generates approximately 2.7-fold more hydroxyl radicals (72% ^•^OH) than superoxide (28% ^•^O_2_^–^),whereas upon light irradiation, it mostly produces superoxide (40% ^•^OH and 60% ^•^O_2_^–^). Upon sonication, **Cy7** forms ~57% of ^•^OH and ~43% ^•^O_2_^–^ but approximately 40% ^•^OH and 60% ^•^O_2_^–^ upon light irradiation.

Importantly, upon sonication and light irradiation, no other radicals except hydroxyl radicals (^•^OH) and superoxide (^•^O_2_^–^) were identified by EPR spectroscopy, and only singlet oxygen (^1^O_2_) was detected among non-radical-based reactive oxygen species by the SOSG-based method (Table 1).

In summary, after sonication, **2ICy7** produces approximately 3.5-fold more radical-based and non-radical-based reactive oxygen species (ROS) than **Cy7**. Among these ROS, ~1.6-fold greater amounts of ^1^O_2_ (Table 1) and ~1.3-fold greater amounts of ^•^OH are generated, but surprisingly ~1.6-fold fewer ^•^O_2_^–^ radicals are formed (Table 2). Upon light irradiation, **2ICy7** forms only approximately 10% more ROS than **Cy7**, which consists of ~3-fold increased concentrations of ^1^O_2_ but approximately the same amount of ^•^OH and ^•^O_2_^–^. Moreover, the relative concentration of radicals and the percentage of light-generated hydroxyl radicals vs. superoxide for **2ICy7** and **Cy7** are less pronounced (Table 2). Importantly, hydroxyl radicals are classified as the most active free radicals; they can react directly with biomolecules and are much more cytotoxic than other ROS, particularly superoxide and singlet oxygen [60,61]. Superoxide radicals are less reactive to biological molecules, but they can cause indirect cell damage. Singlet oxygen has a very short diffusion distance, limiting its direct cytotoxicity [61]. Hydroxyl radicals are, therefore, the most important radicals for PDT and SDT applications, although in practice, all these cytotoxic species can work together. We can anticipate, therefore, that **2ICy7** will show increased sonotoxicity in cells compared to **Cy7**.

### 2.4. Photothermal and Sonothermal Effects

Cytotoxic effects caused by light or US irradiation can be due not only to the generation of singlet oxygen and/or free radicals, but also to heating. The thermal effect is used in photothermal (PTT) and sonothermal (STT) therapies [62]. We investigated this effect by irradiating 20 µM dye solutions in DMSO, similar to previously reported procedures [63]. The dyes were spectrophotometrically proven to not aggregate at these concentrations. The samples at the same initial temperature (25 °C) were light irradiated (730 nm LED, 40 mW/cm^2^) or sonicated (1 MHz) at powers of 0.7 W/cm^2^, 1.2 W/cm^2^, and 2.0 W/cm^2^. Consequently, the time-dependent temperature increase for the dye solutions vs. DMSO (without dye) was measured (Figure 3). Both dyes exhibited almost equal photothermal effects (~7 °C), while the solvent itself was heated by only ~2 °C (Figure 3A). The corresponding thermal curves reached saturation after ~10–12 min of irradiation. Moreover, upon sonication at 0.7 W/cm^2^ and 1.2 W/cm^2^, no thermal effect was detected (Figure 3B). Only after 10 min of sonication at 2.0 W/cm^2^ was a temperature increase of approximately 5 °C detected for both dyes compared with the solvent alone. In our further animal experiments, we applied a power of 0.7 W/cm^2^, at which the sonothermal effect was negligible. In contrast, the light irradiation power in our in vivo PDT experiments was 70 mW/cm^2^, which was higher than that in the photothermal experiment (40 mW/cm^2^, Figure 3A); therefore, light irradiation is expected to induce pronounced photothermal cytotoxicity.

### 2.5. Comparison of Cytotoxic Factors

The relative contributions of the different cytotoxic effects described above are shown in Figure 4. The presented diagrams demonstrate that **2ICy7**, compared to **Cy7**, more efficiently produces singlet oxygen upon light and US irradiation (Figure 4A,B), but also hydroxyl radicals and superoxide upon sonication (Figure 4C), while the photoactivated generation of hydroxyl radicals and superoxide is approximately equal. Remarkably, both dyes exhibit almost the same photothermal effect (~4.5 °C at ~25 J/cm^2^), but no reliable temperature increase under sonication even at ~420 J/cm^2^ (Figure 4D).

In summary, **2ICy7**, compared with **Cy7**, should have stronger phototoxic and sonotoxic effects because of the more efficient production of singlet oxygen. In addition, the higher sonotoxicity of **2ICy7** vs. **Cy7** is due to the more pronounced generation of hydroxyl radicals and superoxide. While singlet oxygen is generated under both photo and US excitations, the mechanisms of the photo- and sonotoxicity of **2ICy7** are suggested to differ to a certain extent. During PDT, the generation of singlet oxygen and thermal effects were observed. In SDT, however, a lack of heating was observed, accompanied by noticeable contributions of hydroxyl radicals and superoxide.

### 2.6. Mouse Tumor Treatment via SDT and Mutual SDT + PDT vs. PDT

We recently investigated PDT with **Cy7–Ab** and **2ICy7–Ab** conjugates in a mouse xenograft model and demonstrated that the introduction of iodine atoms drastically improves the photo-eradication of tumors, with no detectable side effects on healthy organs [41]. In the present study, we investigated the SDT and combined SDT + PDT effects of these conjugates and compared the obtained data with those for the above-described PDT approach. A schematic representation of these experiments is shown in Figure 5.

In brief, 30 human breast cancer-bearing mice (BT-474 cell line, tumor volume ~42 mm^3^) were randomly separated into six groups (5 mice per group). Group 1 was used as the control for tumor growth and background autofluorescence in imaging experiments. Group 2 was used to evaluate the effect of ultrasound alone. Groups 3 and 4 were intravenously (IV) administered (tail) the **2ICy7–Ab** conjugate (100 μg in 200 μL of PBS), and groups 5 and 6 were IV administered the same dose of **Cy7–Ab** to evaluate the effects of SDT and the combination of SDT and PDT. The doses of the conjugates selected for the in vivo study (100 μg) were the same as those used in our previous work [41].

The distribution of the conjugates in the mouse whole body was monitored via fluorescence imaging at 6 h and 24 h post-injection. After 6 h, an intense fluorescence signal originating from the sensitizer was observed, mostly in the lungs. After 24 h, complete accumulation was achieved (the ratio between the bright fluorescence signal in the tumor and the weak signal from the benign tissues exhibited no detectable change; Figure 6). At this time point (24 h), groups 3 and 4 (administered **2ICy7–Ab**) and groups 5 and 6 (administered **Cy7–Ab**) were exposed to ultrasound (US dose 210 J/cm^2^), whereas groups 4 and 6 were subjected to light irradiation. The light dose was 63 J/cm^2^ at 70 mW/cm^2^, which was the same as that used in our previous work [41]. The light power was within the range applied by other researchers (50–100 mW/cm^2^) [64]. Notably, the applied US dose was two times lower than the safe dose (420 W/cm^2^ at 2 W/cm^2^, 1 MHz) previously reported in the literature [57]. The instrument settings for the mouse experiments, including the applied US and light irradiation doses, were the same as those for the above sono- and photostability measurements, which allowed us to estimate the effect of dye degradation on the sensitizing properties (Figure 1). The dyes almost completely decomposed at a light dose of 63 J/cm^2^ (Figure 1A,B); therefore, increasing the applied light dose to improve photodynamic treatment without additional administration of the sensitizer is not reasonable. Moreover, we used a much higher US dose (210 J/cm^2^) at which at least 60% of the Ab-conjugated sensitizers remained undamaged (Figure 1C,D), and this dose could still be increased.

The tumor sizes were monitored for 45 days using the hands-on digital caliper method (Figure 6) [65]. On the 45th day, all the tumors and organs (liver, spleen, lung, kidney, heart, and brain) were resected for further imaging and histopathological analysis.

It was found that tumors in the non-sonicated and sonicated control groups 1 and 2 started to grow gradually after the 5th day of observation, whereas the **Cy7–Ab**-treated tumors in groups 5 and 6 showed modest growth for 13 days and then began to grow rapidly (Figure 7). By the 45th day, the tumors of control groups 1 and 2 reached 821 ± 48 mm^3^ and 766 ± 48 mm^3^, respectively, demonstrating an ~19-fold increase in size compared with the initial volume (~42 mm^3^), whereas the tumors of groups 5 and 6 exhibited an ~12-fold increase (the final tumor volumes were 522 ± 24 mm^3^ and 490 ± 19 mm^3^), which corresponded to moderate ~1.6-fold tumor growth suppression compared with that of group 1. This finding indicated that sonication without the conjugate (group 2) had no detectable effect on the tumors, whereas sonication of **Cy7–Ab**-injected mice slightly suppressed tumor growth, which could be attributed to the therapeutic effect caused by the antibody. A similar moderate effect (~1.4-fold tumor suppression) was recently observed for **Cy7–Ab**-injected light-irradiated and non-irradiated (kept in the dark) mice [41]. Thus, the photo- and sonodynamic effects of **Cy7–Ab** on tumor growth were negligible.

In contrast, the tumors of the US-irradiated (group 3) and US + light-irradiated (group 4) mice that were administered **2Cy7–Ab** initially showed the same trend of growth as those in groups 5 and 6, but one week after irradiation, they started to rapidly decrease. During the next week, the sonicated tumors (group 3) exhibited an ~3.5-fold decrease (to 12 ± 9 mm^3^) in size compared with the initial tumor size (~42 mm^2^), which corresponded to ~68-fold tumor growth suppression compared with that of control group 1 and ~42-fold suppression compared with that of **Cy7–Ab**-treated groups 5 and 6.

Moreover, tumors in the US + light-irradiated group 4 exhibited an ~4.7-fold reduction to 9 ± 5 mm^3^, which corresponded to ~91-fold tumor growth suppression compared with that in control group 1 and ~56-fold suppression compared with that in **Cy7–Ab**-treated groups 5 and 6. These results provide evidence that SDT + PDT with **2ICy7–Ab** (group 4) could be an even more efficient treatment approach, causing almost complete tumor elimination, than sonication alone (group 3), even though the contribution of PDT to SDT + PDT is moderate. The use of PDT in the clinic is, however, limited by its light penetration depth. However, **2ICy7–Ab**-mediated PDT (without SDT) causes only 5.4-fold tumor growth suppression [41], while SDT results in 68-fold tumor suppression (~13 times in our experimental setup) but also in noticeable tumor reduction.

Importantly, the sensitizers can decompose upon US and/or light exposure, as shown in Figure 1. At the applied US dose (210 J/cm^2^, blue stars in Figure 1), the concentration of the **2ICy7–Ab** conjugate decreased to ~47% of the initial value, which means that an even higher US dose can be applied to achieve a better treatment effect after a single IV injection. At the same time, upon light irradiation (a light dose of 63 J/cm^2^, red star in Figure 1), this conjugate almost completely decomposes, and further photodynamic treatment requires an additional injection of **2ICy7–Ab**. Thus, taking into account stronger tumor growth suppression, deeper penetration into the body, and much higher sonostability (*D*_1/2,US_~382 J/cm^2^ by absorption and 349 J/cm^2^ by emission, Table 1) than photostability (*D*_1/2,Light_~9.3 J/cm^2^ by absorption and 7.8 J/cm^2^ by emission), SDT with the **2ICy7–Ab** conjugate is considered a substantially more efficient treatment modality than PDT.

The tumor volumes measured by the caliper method in vivo on the 45th day were consistent with those measured after tumor resection (the volumes of the resected tumors are shown as stars in Figure 7). The resected tumors in the US-irradiated group 2 (864 ± 34 mm^3^) were similar in size to those in control group 1 (834 ± 45 mm^3^). The tumors of groups 5 and 6 were 621 ± 27 mm^3^ and 606 ± 46 mm^3^, respectively, which was approximately 1.4-fold lower than that of group 1, whereas the tumors of groups 3 and 4 were significantly smaller, 15 ± 8 mm^3^ and 9.5 ± 5 mm^3^, respectively. The significant reduction in tumor size for the **2ICy7–Ab**- vs. the **Cy7–Ab**-treated mice upon PDT [41] and, even greater, upon SDT and SDT + PDT (Figure 6 and Figure 8) agrees well with the φ_Δ,US_ and Φ_Δ,Light_ of these conjugates (Table 1).

The body weight of the **2ICy7–Ab**-treated mice was unchanged during the 45 days of investigation, whereas the control and **Cy7–Ab**-treated mice exhibited weight loss (Appendix A).

The resected tumors, livers, spleens, lungs, kidneys, hearts, and brains were captured under white light and then histopathologically analyzed. Notably, a **Cy7–trastuzumab** conjugate was previously evaluated in a breast cancer xenograft for specific accumulation in HER2+ (BT-474) versus HER2–(MDA-MB-231) cell lines [66].

### 2.7. Histopathological Study

The tumor development and side effects of the sonodynamically treated (group 3) and untreated (group 1) mice were analyzed via histopathological analysis of the tumors and organs (liver, spleen, lung, kidney, heart, and brain). The obtained results are presented in Table 3. Histopathological analysis revealed that all five control mice (group 1) had subcutaneous tumors with extensive necrosis. Moreover, four control animals had metastases: two mice had metastases in the liver, and the other two mice had metastases in the lung, spleen, kidney, and peritoneum. Previously, it was reported that BT-474 can metastasize in vivo [67,68]. Interestingly, the liver and kidney metastases created solid masses, whereas the lung metastases showed an interstitial spread of multiple microscopic foci. Peritoneal metastases were observed in tumor cells surrounding and attached to the spleen capsule without penetrating it. Importantly, lymphovascular invasion (LVI) was identified within one of the spleen blood vessels, indicating that this is the mechanism of tumor spread (Figure 9).

In sharp contrast, none of the sonodynamically treated animals (group 3) had metastases. Moreover, two of the treated animals did not have a subcutaneous tumor at the site of tumor implantation (Figure 9). The heart and brain were free of metastases in all the mice in both the control and treated groups (Appendix A). In all the examined tissues (liver, spleen, lung, kidney, heart, and brain), there was no morphological evidence of treatment-related injury, such as nuclear atypia, degenerative changes, inflammation, fibrosis, or vascular alteration. Hence, sonodynamic therapy with an antibody-guided **2ICy7–Ab** sensitizer has no morphologically detectable side effects on the main nontargeted, nontumoral tissues.

## 3. Materials and Methods

### 3.1. Materials

The **Cy7** and **2ICy7** dyes and dye–trastuzumab conjugates **2ICy7–Ab** (dye-to-antibody ratio, DAR~1.8) and **Cy7–Ab** (DAR~1.7) were the same as those used in our previous work [41]. The structure confirmation data are presented in Appendix A. The dyes were >95% pure according to HPLC. All other chemicals were obtained from Alfa Aesar (Petach Tikva, Israel) and Sigma-Aldrich (Jerusalem, Israel). Solvents were purchased from Bio-Lab (Jerusalem, Israel) and used as is.

### 3.2. Photostability and Sonostability

Solutions of the dyes and conjugates (*c*~1 μM) in 0.1 M PBS, pH 7.4, were prepared. For the photostability measurements, these solutions were irradiated from a distance of 26.5 cm in a standard 1 cm quartz cell by using a 730 nm, 30 W LED equipped with a 60° lens; the light irradiance was 8 mW/cm^2^.

Sonication in all experiments was carried out using an ultrasound therapy unit (Dr. Equipment, New Delhi, India). For the sonostability measurements, the sample solutions were sonicated (*f* = 1 MHz, 0.7 W/cm^2^) by a US transducer (diameter 30 mm) immersed at a depth of 1 mm on the upper surface of the solutions (5 mL) placed in 55 mm glass Petri dishes. The absorption and emission spectra of the solution were recorded over time in a standard 1 cm quartz cell, and plots of the normalized absorbance and fluorescence intensity vs. time were drawn. Then, the photo- and sonostabilities of the dyes were quantified through the half-lives (τ_1/2_) calculated from the corresponding monoexponential absorption and fluorescence decay functions (Equation (1)).
*I* = *I*_0_ + *A* × exp(–*τ*/*t*),(1)
where *I* is the time-dependent absorbance or fluorescence intensity, *I*_0_ is the offset of intensity, *A* is the amplitude, *t* is the decay constant, and τ is time.

### 3.3. Quantum Yields of Photodynamic and Sonodynamic Singlet Oxygen Generation

The quantum yields of the photodynamic singlet oxygen generation (Φ_Δ,Light_) were measured in 0.1 M PBS, pH 7.4, according to a previous procedure [42,69] using singlet oxygen sensor green (**SOSG**, *c* = 6 µM) (Thermo Fisher Scientific Inc., Waltham, MA, USA) as the singlet oxygen indicator.

The relative quantum yields of singlet oxygen generation under US exposure (φ_Δ,US_) were measured in 0.1 M PBS, pH 7.4, similar to the reported procedure [69]. Solutions of **SOSG** (*c* = 6 µM) [70] and the dye under investigation or the reference dye (**Cy7**) (*c* = 1.7–2.3 µM) in PBS (5 mL) were prepared and sonicated at *f* = 1 MHz, and the applied US intensity was 2 W/cm^2^. The emission spectra of the solutions were recorded over time in standard 1 cm quartz cells (λ* = 488 nm). The total exposure time reached 8 min. During this time, the emission of **SOSG** at 530 nm gradually increased. The corresponding plots representing the emission intensity of **SOSG** versus time were plotted and fitted by a zero-order reaction rate function [69], the rates of singlet oxygen scavenger degradation (*r*) were calculated (Appendix A), and the relative φ_Δ,US_ values of the dyes and conjugates were quantified via Equation (2) [69], with **Cy7** in PBS (φ_Δ,US,Ref_ ≡ 1 a.u.) used as the reference.
φ_Δ_ = φ_Δ,US,Ref_ × (*r*/*r*_Ref_) × (*A*_Ref_/*A*),(2)
where φ_Δ,US,Ref_ is the relative sonodynamic quantum yield of the singlet oxygen generation for the reference dye (1 a.u.), *r*_Ref_ and *r* are the singlet oxygen scavenger degradation rates obtained from the corresponding fitting curves of the reference dye and the dye under examination, respectively, and *A*_Ref_ and *A* are the absorbances at λ = 750 nm of the reference dye (**Cy7**) and the dye or conjugate under examination, respectively.

Each φ_Δ,US_ was measured three times, and the average value was taken; the reproducibility was within 5%.

### 3.4. Detection of Free Radicals by EPR

EPR measurements were carried out at room temperature using a Bruker ELEXYS E500 spectrometer operating at X-band frequencies (9.5 GHz) and a Bruker ER4119HS resonator. Solutions of 20 μM **Cy7** and **2ICy7** with a 25 mM BMPO spin trap in double-distilled water were prepared. The samples were dissolved in 1 mL of spin trap solution and loaded into a Vitrocom quartz capillary (CV1012-Q-100) with a 1 mm inner diameter.

To measure US-generated radicals, sonication was carried out for 15 min using an MRC ACP-120H bath sonicator (40 kHz, 50 W) (MRC Lab, Holon, Israel), and then the EPR spectra were recorded.

For determination of photogenerated radicals, the samples were measured for 30 min upon light irradiation (730 nm, 30 W LED, light power density of 8 mW/cm^2^, light dose of 14.4 J/cm^2^).

The experimental conditions for the EPR measurements were as follows: a microwave power of 20 mW, a 1 Gauss modulation amplitude, and a 100 kHz modulation frequency; the sweep range was 150 Gauss; and the spectra consisted of 300 points. The data were plotted using Microcal Origin software, version 8.6 (OriginLab Corp., Northampton, MA, USA).

For the quantification of radicals, the signal intensity was compared against a linear-fit calibration curve obtained with known concentrations (10 μM, 20 μM, 40 μM, and 80 μM) of 3-carboxy-PROXYL. The intensities of total BMPO-bound radicals, BMPO-OH radicals, and BMPO-OOH were quantified by double integration of the spectra, and the concentrations were calculated using a calibration curve.

### 3.5. Photothermal Measurements

Solutions (8 mL) of **Cy7** and **2ICy7** in DMSO (20 µM) and DMSO (as a control) were placed in non-covered 60 × 12 mm glass Petri dishes and light irradiated (730 nm, 30 W LED, 40 mW/cm^2^) from the top, perpendicular to the surface of the liquid, from a distance of 10 cm. The initial temperature for all the solutions before irradiation was the same at 25 °C, and the temperature increase was simultaneously recorded upon irradiation via a thermal imager FLUKE TiS20 (Fluke Corp, Everett, WA, USA) positioned at the top of the solutions at an angle of 45°. The graphs for the temperature increase (Δ*t* vs. time) were subsequently plotted.

### 3.6. Sonothermal Measurements

The measurements were carried out similarly to those of the photothermal experiment, but the irradiation was performed using an ultrasound therapy unit (Dr. Equipment, New Delhi, India) at 0.7 W/cm^2^, 1.2 W/cm^2^, and 2.0 W/cm^2^ (*f* = 1 MHz). The solutions (3 mL) were placed directly on top of an ultrasound transducer (diameter 3.5 cm) equipped with a plastic border, and the temperature was measured from the top of the solutions.

Each photothermal and sonothermal experiment was performed in triplicate, and the average values were taken.

### 3.7. Mouse Preparation

All experiments with mice were carried out in compliance with the Israel Council on Animal Care regulations and were approved by the Animal Care Committee of Ariel University (authorization number IL-179-06-19, 21 May 2023).

Thirty-six-week-old athymic BALB/c female nude mice (Harlan Labs, Nes Ziona, Israel) were subcutaneously inoculated on the dorsal right side with the human breast cancer cell line BT-474 (1 × 10^6^ cells in PBS into nu/nu mice, 100 µL per mouse), and tumors were allowed to establish until the tumor sizes reached approximately 42 mm^3^ (16 days). The tumor-bearing mice were then randomly separated into six groups (5 mice per group). Group 1 was used as the control for tumor growth and background autofluorescence in imaging experiments. Group 2 was used to evaluate the effect of ultrasound alone. Groups 3 and 4 were intravenously (IV) administered (tail) the **2ICy7–Ab** conjugate (100 μg in 200 μL of PBS), and groups 5 and 6 were IV administered **Cy7–Ab** at the same dosage to evaluate the sonodynamic (SDT) and combined sono- (SDT) and photodynamic (PDT) effects.

### 3.8. Anesthesia

Before imaging and/or photo/sonodynamic treatment, the mice were anesthetized via an intraperitoneal injection of a combination of ketamine and xylazine. Dose: 0.5 mL of ketamine + 0.25 mL of xylazine + 4.25 mL of water for injection. The dose rate was 0.1 mL/10 g of body weight.

### 3.9. Animal Imaging

Imaging was carried out using an IVIS Spectrum multispectral fluorescence in vivo imaging system (PerkinElmer, Waltham, MA, USA). For imaging, the mice were anesthetized, and imaging was performed after conjugate administration at 6 h and 24 h post-injection. The images were obtained in white light mode with the following fluorescence channel: 710/30 nm excitation bandpass filter and 780/20 nm emission bandpass filter. The data were processed with IVIS Living Image software, version 4.8.2.

### 3.10. PDT and SDT Experiments

The BT-474 tumor-bearing mice were established as described above until the tumor volume reached approximately 42 mm^3^. The mice were randomly divided into six groups. Group 1 was used as the control for tumor growth and background autofluorescence in imaging experiments. Group 2 was used to evaluate the effect of ultrasound alone. Groups 3 and 4 were administered **2ICy7–Ab** via the tail vein (IV)**,** and groups 5 and 6 were administered **Cy7–Ab** (100 µg in 200 μL of PBS). The accumulation of the conjugates was monitored by fluorescence imaging at 6 h and 24 h. The accumulation of this conjugate is known to be achieved after 24 h [41]. At this time point, the mice in groups 2–6 were anesthetized and subjected to ultrasound using the ultrasonic device described above. The irradiation conditions were as follows: transducer diameter, 30 mm; gel, *f* = 1 MHz; applied ultrasound intensity, 0.7 W/cm^2^; 1 min irradiation/1 min rest, 5 times; and overall US dose, 210 J/cm^2^. This US dose and sonication regime was selected on the basis of the behavior of the mice. Increasing power and/or dose resulted in anxiety and distress in the mice.

Additionally, groups 4 and 6 were subjected to NIR light irradiation (730 nm, 30 W LED, light power density 70 mW/cm^2^, 15 min, light dose 63 J/cm^2^). Then, the tumor sizes were measured in vivo over time using the hands-on digital caliper method, and the tumor volumes were calculated according to Equation (3) [65] and are presented on the plot as the mean ± standard error of the mean (SEM).
Tumor volume (mm^3^) = 0.5 × Tumor Length × Tumor Width^2^(3)

### 3.11. Histological Protocol

The tumor-bearing mice were sacrificed, and the tumors and organs (liver, spleen, lung, kidney, heart, and brain) were harvested and fixed in formalin to prepare paraffin sections. Hematoxylin/eosin (H&E) staining was used for histological analysis.

### 3.12. Pathological Evaluation of H&E-Stained Sections

The study pathologist was blinded to the experimental protocol used for each animal. H&E-stained sections of the tumor and organs were examined for any morphological abnormalities. The amount of necrosis in the tumor tissues was estimated as a percentage of the total tumor area.

## 4. Conclusions

In this study, we explored the efficacy of a trastuzumab-guided diiodinated heptamethine cyanine-based sensitizer, **2ICy7–Ab**, for sonodynamic therapy (SDT) in an HER2+ xenograft model. The sonosensitizing effect was compared to that of the noniodinated counterpart, **Cy7–Ab**. Investigations have also been conducted on the combined sonodynamic and photodynamic (PDT) effects. The sonodynamic impact of **2ICy7–Ab** in tumor treatment (68-fold tumor growth suppression vs. control) was much more pronounced (~43 times) than that of **Cy7–Ab** (1.6-fold suppression). Upon combined SDT and PDT, the sonodynamic outcome noticeably surpassed that of the photodynamic treatment alone. After a single intravenous **2ICy7–Ab** injection, SDT resulted in a 3.5-fold reduction in tumor volume within 45 days and exhibited 13-fold greater tumor suppression than PDT. Moreover, during this period, SDT with **2ICy7–Ab** did not cause any observable (weight, physical conditions, or complete survival) toxic effects in the animals, or morphologically detectable side effects on the non-targeted tissues or organs. The evaluation of the potential long-term side effects of the treatment will be addressed in our future studies.

The iodine atoms in the **2ICy7** dye were identified as pivotal contributors to the enhanced photo- and sonotoxicity of the **2ICy7–Ab** conjugate, as the non-iodinated **Cy7–Ab** analog exhibited no statistically significant cytotoxicity toward tumors. Upon light and ultrasound irradiation, the iodinated **2ICy7** dye and the **2ICy7–Ab** conjugate demonstrated noticeably increased quantum yields of singlet oxygen generation compared with their non-iodinated counterparts. Additionally, the higher sonotoxicity of the iodinated versus the non-iodinated dyes is driven by the increase in hydroxyl radical and superoxide generation.

Crucially, **2ICy7–Ab** provides a sufficiently strong fluorescent signal, facilitating real-time tracking of the distribution of the sensitizer throughout the body and its accumulation in the tumor. This capability is essential for identifying the optimal time for sonication, thereby enhancing the precision and safety of the treatment. Given that SDT can be applied to the whole body and multiple organs [71] and that it is a deeper treatment approach than PDT, we assume that SDT with the **2ICy7–Ab** conjugate is, in general, a more promising therapeutic approach than PDT. However, determining the localization of the conjugate in host organs via fluorescence imaging prior to treatment is important. Our future study will include an investigation of the impact of the number of iodines, their positioning over the dye scaffold on RS generation, and the efficacy of sensitizer-antibody conjugates on various HER2 overexpressed and HER2 low-expressed tumors in vivo. The limitations associated with the DAR range, aggregativity, solubility, and bystander effect of our conjugates will also be explored. Another limitation relates to the heterogeneity of spontaneously arising tumors [72]. The further development of the proposed approach to address this issue is challenging.

## 5. Patents

A patent on iodinated cyanine–antibody conjugates for sonodynamic applications is pending.

## Figures and Tables

**Figure 1 ijms-25-10137-f001:**
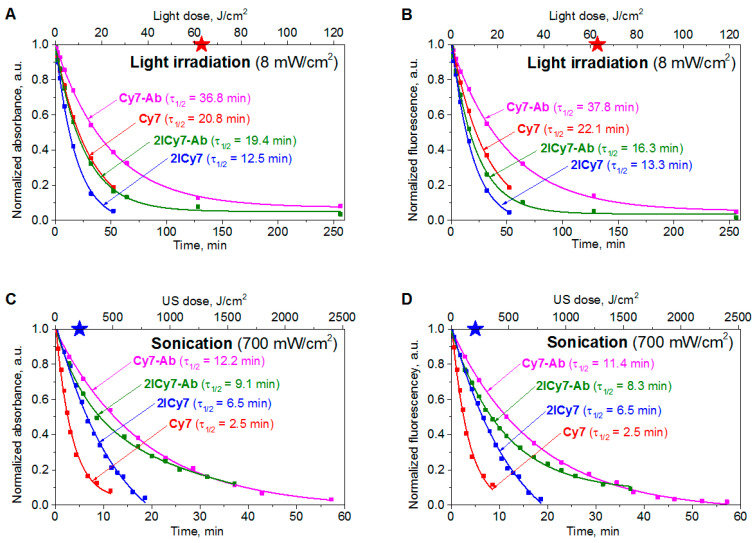
The normalized absorption (**A**,**C**) and fluorescence intensities (**B**,**D**) of **Cy7**, **2ICy7**, **Cy7–Ab**, and **2ICy7–Ab** (*c* = 1 μM) upon light (**A**,**B**) and US (**C**,**D**) irradiation in 0.1 M PBS, pH 7.4. Light irradiation: 730 nm light-emitting diode (LED), power density 8 mW/cm^2^. Sonication: frequency (*f*) 1 MHz, 0.7 W/cm^2^. The red and blue stars show the irradiation light (63 J/cm^2^) and US (210 J/cm^2^) doses applied for PDT and SDT, respectively, in the mouse model.

**Figure 2 ijms-25-10137-f002:**
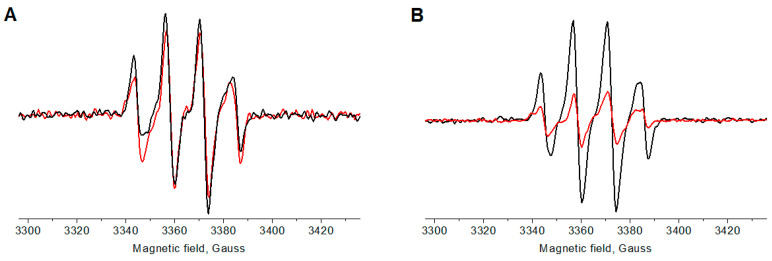
Electron paramagnetic resonance (EPR) spectra of 2ICy7 (black traces) and Cy7 (red traces) (dye concentration cDye = 20 µM) upon 30 min of light irradiation (**A**) and after 15 min of sonication (**B**).

**Figure 3 ijms-25-10137-f003:**
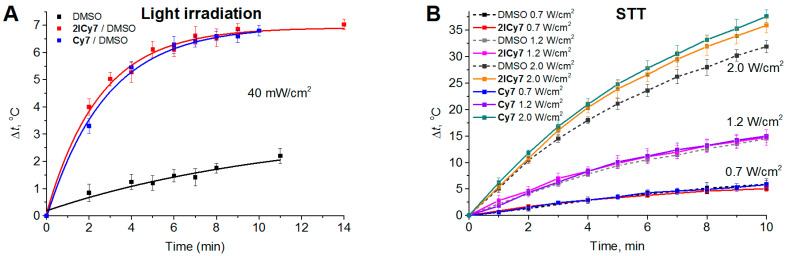
Temperature increase for the **Cy7** and **2ICy7** solutions in DMSO (*c*_Dye_ = 20 µM) vs. dimethyl sulfoxide (DMSO) upon light irradiation (**A**) and sonication (**B**). Light irradiation: 730 nm LED, 40 mW/cm^2^. Sonication: 1 MHz, 0.7 W/cm^2^, 1.2 W/cm^2^, and 2.0 W/cm^2^.

**Figure 4 ijms-25-10137-f004:**
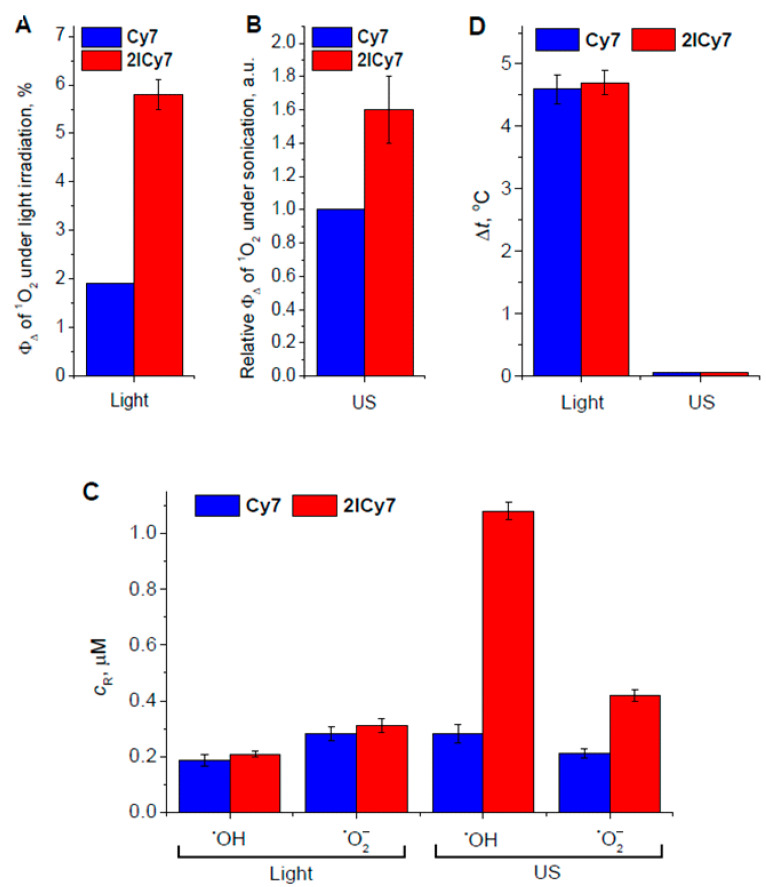
Quantum yield of singlet oxygen (^1^O_2_) upon light irradiation (**A**) and sonication (**B**); relative concentrations of hydroxyl radicals (^•^OH) and superoxide (^•^O_2_^–^) (**C**); photothermal and sonothermal effects (**D**). *c*_R_ is the concentration of free radicals; Φ_Δ_ is the quantum yield of ^1^O_2_ generation.

**Figure 5 ijms-25-10137-f005:**
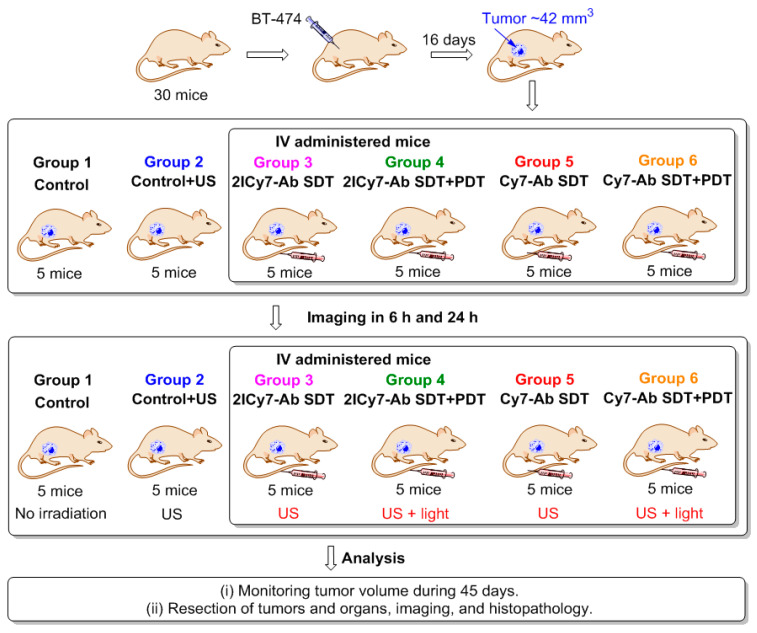
Mouse experiment. Thirty mice were inoculated with BT-474 cells. Group 1 was used as the control for tumor growth and background autofluorescence upon imaging. Group 2 was used to evaluate the effect of ultrasound alone. After 15 days, the mice were intravenously (IV) administered (tail) **2ICy7–Ab** (groups 3 and 4) or **Cy7–Ab** (groups 5 and 6). The administered dose was 100 μg (5 mg/kg) of conjugate in 200 μL of PBS. Twenty-four hours post-injection, the mice were exposed to US (SDT, *f* = 1 MHz, 0.7 W/cm^2^, 1 min irradiation/1 min rest, 5 times) (groups 2–6) followed by NIR light irradiation (PDT, groups 4 and 6 only) for 15 min (730 nm LED, 70 mW/cm^2^, 63 J/cm^2^). The tumor volumes for all the groups were subsequently monitored for 45 days. Tumors and organs were then resected and analyzed by imaging. PDT is photodynamic therapy; SDT is sonodynamic therapy; US is ultrasound.

**Figure 6 ijms-25-10137-f006:**
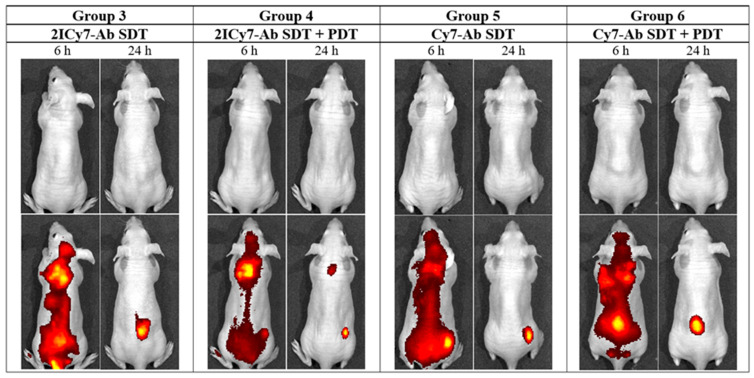
In vivo whole-body images of representative tumor-bearing mice (groups 3–6) captured at 6 h and 24 h after intravenous (IV) injection of the Cy7–Ab and 2ICy7–Ab conjugates. 1st row: white light; 2nd row: overlay of the fluorescence channel with white light. PDT is photodynamic therapy; SDT is sonodynamic therapy.

**Figure 7 ijms-25-10137-f007:**
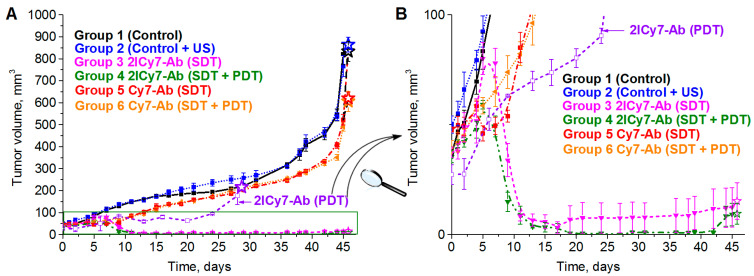
Tumor growth curves ((**A**), full range and (**B**), zoom out) of BT-474 tumor-bearing mice: control (group 1), monitored after ultrasound (group 2), monitored after intravenous (IV) administration of 100 µg (0.5 mg/mL) of **2ICy7–Ab** or **Cy7–Ab**, when irradiated with US (SDT, f = 1 MHz, 0.7 W/cm^2^, 1 min irradiation/1 min rest, 5 times) (groups 3 and 5), and **2ICy7–Ab** or **Cy7–Ab**, when irradiated with US (SDT, the same conditions) and subsequently exposed to light (PDT, 730 nm LED, 63 J/cm^2^) for 30 min (groups 4 and 6). For comparison, the tumor volumes for the mice irradiated with light alone (PDT, 730 nm LED, 63 J/cm^2^) [41] are also shown (violet curve 2ICy7–Ab (PDT)). The tumor volumes were measured in vivo using the caliper method. The volumes of the resected tumors (see Figure 6) are shown by stars. The tumor volume at each time point is represented by the mean ± SEM for five mice in each group. PDT is photodynamic therapy; SDT is sonodynamic therapy.

**Figure 8 ijms-25-10137-f008:**
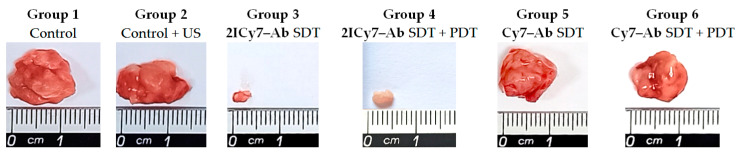
Representative photographs of tumors resected on the 45th day after intravenous (IV) injection of the conjugates. Non-administered sonicated (group 1) and non-sonicated (group 2) mice were used as controls. PDT is photodynamic therapy; SDT is sonodynamic therapy.

**Figure 9 ijms-25-10137-f009:**
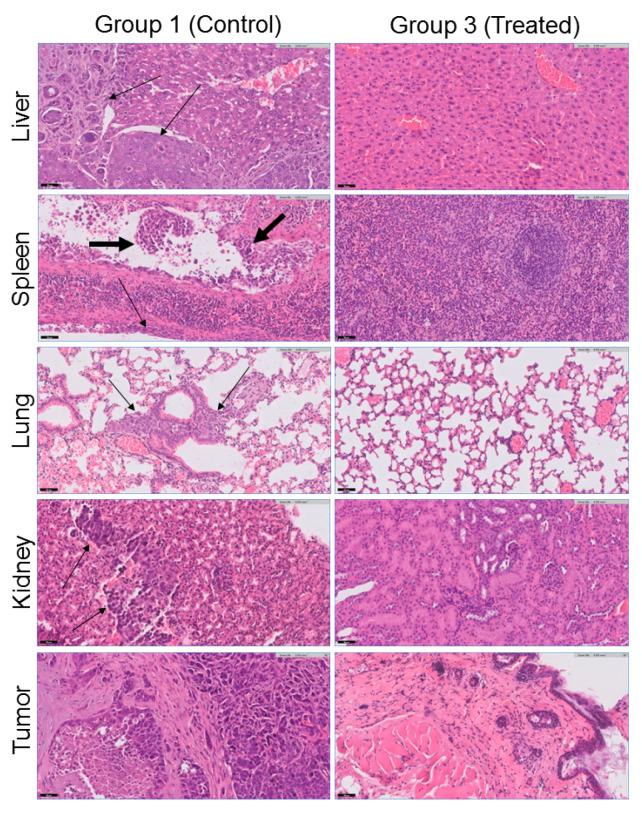
Representative histopathological findings (hematoxylin/eosin (H&E) staining, magnification ×20) showing differences between the control (group 1) and SDT-treated (group 3) mice. Metastases (marked by thin arrows) were identified in four of five control animals in the liver (image for mouse #1 from group 1 is shown, see Table 3), spleen, peritoneum, lung, and kidney (image for mouse #3 is shown). The liver and kidney metastases created solid masses, while the lung metastases showed an interstitial spread of multiple microscopic foci. LVI was identified within one of the spleen blood vessels, indicating that this is the mechanism of tumor spread, marked by a thick arrow. All control animals (group 1) had a subcutaneous tumor, whereas two of the treated animals (group 3) had no evidence of a tumor at the site of implantation.

**Table 1 ijms-25-10137-t001:** The degradation half-lives of the dyes and conjugates with different dye-to-antibody ratios (DAR) (*c* = 1 μM) upon light (τ_1/2,Light_) and US (τ_1/2,US_) irradiation, obtained from the absorption/fluorescence (Ab/Fl) intensities, corresponding light (*D*_1/2,Light_) and US (*D*_1/2,US_) doses, and the quantum yields of photodynamic (Φ_Δ,Light_) and sonodynamic (φ_Δ,US_) singlet oxygen generation, measured in 0.1 M PBS, pH 7.4.

Dye	τ_1/2,Light_, min at 8 mW/cm^2^ (Ab/Fl)	τ_1/2,US_, min at 0.7 W/cm^2^ (Ab/Fl)	*D*_1/2,Light_, J/cm^2^ at 8 mW/cm^2^ (Ab/Fl)	*D*_1/2,US_, J/cm^2^ at 0.7 W/cm^2^ (Ab/Fl)	Φ_Δ,Light_, % [41]	φ_Δ,US_, a.u.
**Cy7**	20.8/22.1	2.5/2.5	10.0/10.6	105/105	1.9	1 *^a^*
**2ICy7**	12.5/13.3	6.5/6.5	6.0/6.4	273/273	5.8 ± 0.3	1.6 ± 0.2
**Cy7–Ab** (DAR 1.7)	36.8/37.8	12.2/11.4	17.7/18.1	512/479	1.1 ± 0.1	3.2 ± 0.2
**2ICy7–Ab** (DAR 1.8)	19.4/16.3	9.1/8.3	9.3/7.8	382/349	3.5 ± 0.3	6.7 ± 0.7

*^a^* **Cy7** was taken as the reference (φ_Δ,US_ ≡ 1 a.u.).

**Table 2 ijms-25-10137-t002:** Concentration (*c*_R_, μM) and percentage of generated radicals *^a^*.

Sample	Light Irradiation			Sonication		
	*c*_R_, μM	^•^OH	^•^O_2_^–^	*c*_R_, μM	^•^OH	^•^O_2_^–^
**2ICy7**	0.520	40%	60%	1.500	72%	28%
**2ICy7** + DMSO	0.312			0.424		
**Cy7**	0.471	40%	60%	0.493	57%	43%
**Cy7** + DMSO	0.282			0.212		

*^a^* Concentrations of radicals (*c*_Rs_) generated upon sonication cannot be compared to those after light irradiation because the applied irradiation doses were different. No other radicals except ^•^OH and ^•^O_2_^–^ were detected by EPR.

**Table 3 ijms-25-10137-t003:** Summary of histopathological results.

Study Group	Group 1 (Control, 5 Mice)	Group 3 (Treated, 5 Mice)
Subcutaneous tumor	A tumor showing necrosis (all 5 mice)	No tumor in 2 out of 5 animals
Liver	Metastasis in liver parenchyma in 2 out of 5 animals (mice #1, #2)	Normal
Spleen	Metastatic peritoneal tumor attached to the spleen capsule and lymphovascular invasion (LVI) of the tumor in the spleen in 2 out of 5 animals (mice #3, #4)	Normal
Lung	Metastatic microscopic foci of interstitial spread in 2 out of 5 animals (mice #3, #4)	Without significant changes
Kidney	Metastasis in kidney parenchyma in 2 out of 5 animals (mice #3, #4)	Normal
Heart	Normal (all 5 mice)	Normal
Brain	Normal (all 5 mice)	Normal

## Data Availability

Dataset available on request from the authors.

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
