# Peer review of "Sonodynamic Therapy for HER2+ Breast Cancer with Iodinated Heptamethine Cyanine–Trastuzumab Conjugate"

_ijms, 2024, doi:10.3390/ijms251810137_

Round 1
Reviewer 1 Report
Comments and Suggestions for Authors
This report relates to determinants of success in eradication of malignant cell types with ultrasound, with and without the addition of light. The authors use the term ‘abnormal’ when they mean ‘malignant’. Ultrasound has a deeper depth of penetration into tissues, compared with light. This could potentiate tumor eradication, assuming that damage to normal host tissues will be minimal.
Use of the term ‘immunotherapy’ apparently relates to the targeting provided by trastuzumab. It is indicated in the text (line 72) that there is a stimulation of ‘the immune system’. The BT-474 cell line was isolated from a patient with ductal breast carcinoma. This can only be implanted in immunosuppressed mice. This issue is not mentioned in the text. Discussing ‘immunotherapy’ in an immunosuppressed animal is a precarious argument. Why was a murine tumor not used? The authors need to explain this.
Agents used can be degraded by either light at 730 nm or ultrasound. It is suggested (Fig. 1) that irradiation, in a mouse model, could be sufficient to cause substantial degradation of agents, with the ultrasound dose somewhat less damaging. This assumes that the doses delivered in a cell-free system will be applicable to an in vivo situation. Formation of OH radical, singlet oxygen and superoxide radical were detected. Thermal effects were also identified.
What does ‘complete accumulation’ mean (line 287)? Fig. 7 indicates maximum efficacy for groups 3 and 4. All figure legends need to be self-explanatory, so a definition of SIT and PIT should be incorporated into the figure legend. Based on the error bars, it appears that adding light to an ultrasound protocol does not create a significant difference. There are no studies involving light + photosensitizer. If these are relatively large tumors, it is likely that light + photosensitizer would have a lesser effect. While the authors can speculate (line 596) on other applications for this approach, treatment of systemic diseases with a protocol involving application of light and/or ultrasound is highly problematic. Where would one direct the light/ultrasound?
Author Response
Reviewer 1
General response to Reviewer 1: We wish to thank this reviewer for his valuable comments to improve our manuscript. The presentation of the results and conclusions is modified according to the reviewer's comments as stated below. The line numeration relates to the track-mode version.
Reviewer 1: This report relates to determinants of success in the eradication of malignant cell types with ultrasound, with and without the addition of light. The authors use the term ‘abnormal’ when they mean ‘malignant’. Ultrasound has a deeper depth of penetration into tissues, compared with light. This could potentiate tumor eradication, assuming that damage to normal host tissues will be minimal.
Authors: We agree with this comment. The term “abnormal” is changed to “malignant” (lines 46 and 72).
Reviewer 1: Use of the term ‘immunotherapy’ apparently relates to the targeting provided by trastuzumab. It is indicated in the text (line 72) that there is a stimulation of ‘the immune system’. The BT-474 cell line was isolated from a patient with ductal breast carcinoma. This can only be implanted in immunosuppressed mice. This issue is not mentioned in the text. Discussing ‘immunotherapy’ in an immunosuppressed animal is a precarious argument. Why was a murine tumor not used? The authors need to explain this.
Authors: We absolutely agree with this comment. The term “immunotherapy”, which is associated with murine tumor models, is removed throughout the text. The text related to the line 72 (now, this are lines 75-81) is also removed. We use human-derived xenografts and therefore immunosuppressed animals (NUDE mice) were applied. To avoid misunderstanding, we substituted the terms "PIT" and "SIT" with "PDT" and "SDT". The immune-related evaluation is not in the scope of this study. Our research focused on the investigation of SDT and its comparison with PDT.
Reviewer 1: Agents used can be degraded by either light at 730 nm or ultrasound. It is suggested (Fig. 1) that irradiation, in a mouse model, could be sufficient to cause substantial degradation of agents, with the ultrasound dose somewhat less damaging. This assumes that the doses delivered in a cell-free system will be applicable to an in vivo situation. Formation of OH radical, singlet oxygen, and superoxide radical were detected. Thermal effects were also identified.
Authors: The generation of different RSs in vivo can differ from that in a cell-free medium, but we assume that this difference will not be pronounced. The investigation of the RS generation in vivo / ex vivo is in our plans.
Reviewer 1: What does ‘complete accumulation’ mean (line 287)? Fig. 7 indicates maximum efficacy for groups 3 and 4. All figure legends need to be self-explanatory, so a definition of SIT and PIT should be incorporated into the figure legend. Based on the error bars, it appears that adding light to an ultrasound protocol does not create a significant difference. There are no studies involving light + photosensitizer. If these are relatively large tumors, it is likely that light + photosensitizer would have a lesser effect. While the authors can speculate (line 596) on other applications for this approach, the treatment of systemic diseases with a protocol involving the application of light and/or ultrasound is highly problematic. Where would one direct the light/ultrasound?
Authors: The term “complete accumulation” means that the ratio between the bright fluorescence signal in the tumor and the week signal from the benign tissues exhibited no detectable change after 24 h post-injection. The corresponding explanation is added (lines 295-298).
“SIT and PIT” was substituted with “SDT and PDT” and incorporated into the Fig. 7 legend.
According to our previous research [41 - https://doi.org/10.1016/j.dyepig.2023.111101], the non-antibody-guided dyes Cy7 and 2ICy7 did not accumulate in tumors sufficiently compared to benign tissues, and therefore we did not irradiate these mice.
SDT on the relatively large tumors can really have a lesser effect, but for large tumors, we can apply higher doses of sensitizers and at least up to 3-4-fold higher irradiation doses per injection. Another treatment regime, e.g., twice a week injection/irradiation, can also be applicable.
We agree with this comment (old line 596), and the text is modified accordingly (lines 615-626). The results of our research clearly pointed to the very moderate contribution of PDT to the efficacy of combined PDT+SDT. Given the fact that SDT can be applied to the whole body and multiple organs (e.g. [71 - https://saisei-mirai.or.jp/en/sonophotodynamic_therapy/]) and that it is a deeper treating approach compared to PDT, we assume that SDT is, in general, more promising than PDT. The corresponding text is modified, and the citation is added (lines 615-626).

Reviewer 2 Report
Comments and Suggestions for Authors
The article titled "Sonoimmunotherapy of HER2+ Breast Cancer with Iodinated Heptamethine Cyanine–Trastuzumab Conjugate" by Dmytro Kobzev et al. provides valuable insights into cancer treatment; however, there are several areas that require attention:
1. The authors should clearly state the aim of the study and the conclusion in the abstract for better clarity and understanding.
2. The optimal dosage or treatment schedule for the cyanine dye-antibody conjugate was not provided by the authors.
3. An in-depth analysis of the underlying mechanisms involved in the increased generation of hydroxyl radicals, superoxide, and singlet oxygen in the iodinated vs. noniodinated counterparts is also required to fully understand the observed sonotoxicity.
4. The study only investigates the treatment of HER2+ human breast cancer, and it would be beneficial to compare the results of studies with other types of cancer (such as luminal and TNBC) to confirm the efficacy of the treatment.
5. It is unclear whether the tumor suppression observed in the study is sustained over a longer period of time or if there are any potential long-term side effects of the treatment that should be addressed in future studies.
6. A detailed analysis of the cost-effectiveness of the treatment should be included in future research.
7. While the study reports no detectable toxicity to off-target tissues, it is possible that there are other side effects that were not investigated in the study, and this should be considered in future research.
8. The authors could include a discussion of the limitations and future perspectives of the study to provide a more comprehensive analysis.
Comments on the Quality of English LanguageModerate editing of English language required.
Author Response
Reviewer 2
General response to Reviewer 2: We wish to thank this reviewer for his valuable comments to improve our manuscript. The abstract, discussion, experimental, and concussions are improved according to the reviewer's comments, as stated below. The line numeration relates to the track-mode version.
The article titled "Sonoimmunotherapy of HER2+ Breast Cancer with Iodinated Heptamethine Cyanine–Trastuzumab Conjugate" by Dmytro Kobzev et al. provides valuable insights into cancer treatment; however, there are several areas that require attention:
Reviewer 2: 1. The authors should clearly state the aim of the study and the conclusion in the abstract for better clarity and understanding.
Authors: The aim (lines 18-21) and the main conclusions (lines 23-35) are underlined in the abstract.
Reviewer 2: 2. The optimal dosage or treatment schedule for the cyanine dye-antibody conjugate was not provided by the authors.
Authors: We are grateful to this reviewer for this valuable comment. The optimal dosage of the conjugates selected for the in vivo study (100 μg) was the same as in our previous work [41 - https://doi.org/10.1016/j.dyepig.2023.111101] (lines 290-292). In the same work, the optimal light dosage (63 J/cm2 at 70 mW/cm2) was found, and we applied this dosage in our current light-irradiation experiments. Regarding SDT, the optimal US dosage (210 J/cm2) and the sonication regime (0.7 W/cm2; 1 min irradiation/1 min rest, 5 times) were found based on mice's behavior. The increasing power and/or dose resulted in mice's anxiety and distress. This explanation is added to the Experimental (lines 565-567).
Reviewer 2: 3. An in-depth analysis of the underlying mechanisms involved in the increased generation of hydroxyl radicals, superoxide, and singlet oxygen in the iodinated vs. noniodinated counterparts is also required to fully understand the observed sonotoxicity.
Authors: The mechanisms of reactive species generation upon sonication are extensively reported in the literature, e.g., [13 - https://doi.org/10.2174/1381612825666190123114107; 39 - https://doi.org/10.1002/chem.201904306]. According to these mechanisms, the increasing efficacy of iodinated dyes is explained by the heavy atom effect, resulting in the increasing triplet state population (see also [40 - https://doi.org/10.1016/j.dyepig.2022.111053; 41 - https://doi.org/10.1016/j.dyepig.2023.111101; 42 - https://doi.org/10.1039/C6CC09624G; 43 - https://doi.org/10.1021/acsami.9b07694; 46 - https://doi.org/10.1016/j.dyepig.2021.109745; 47 - https://doi.org/10.1016/j.dyepig.2020.108854]) (lines 45-53). We agree that a dipper analysis of these effects is needed for the dyes explored in our work, and such research is planned to be carried out in our lab.
Reviewer 2: 4. The study only investigates the treatment of HER2+ human breast cancer, and it would be beneficial to compare the results of studies with other types of cancer (such as luminal and TNBC) to confirm the efficacy of the treatment.
Authors: We absolutely agree with this comment. It is valuable to test these dye-trastuzumab conjugates on Her2 low-expressed tumors like TNBC. Giving the fact that another trastuzumab-based ADC, Enhertu, is effective in such tumors testing our sono-ADC is rational. However, such experiments require a new cohort of animals and are time-consuming. Such research is also planned to be performed in our lab.
Reviewer 2: 5. It is unclear whether the tumor suppression observed in the study is sustained over a longer period of time or if there are any potential long-term side effects of the treatment that should be addressed in future studies.
Authors: The purpose of our research was to compare SDT/PDT/SDT+PDT and iodinated vs. noniodinated conjugates. We achieved satisfactory results within 45 days. During this period, we did not observe any systemic toxic effects with the iodinated conjugate. We agree with this reviewer: the potential long-term side effects of the treatment will be addressed in future studies. The corresponding text is added to the conclusions (lines 602-603).
Reviewer 2: 6. A detailed analysis of the cost-effectiveness of the treatment should be included in future research.
Authors: We absolutely agree with this comment. The cost-effectiveness of the treatment will be investigated in our future research.
Reviewer 2: 7. While the study reports no detectable toxicity to off-target tissues, it is possible that there are other side effects that were not investigated in the study, and this should be considered in future research.
Authors: We plan: MTD, PK, and repeated dose (safety) study on mice and rats. Such a study is scheduled exhaustive toxicology. See also lines 598-603 (side effects).
Reviewer 2: 8. The authors could include a discussion of the limitations and future perspectives of the study to provide a more comprehensive analysis.
Authors: The corresponding discussion is added: "Our future study will include the investigation of the impact of the number of iodines and their positioning over the dye scaffold on the RS generation and efficacy of sensitizer-antibody conjugates on various Her2 overexpressed and Her2 low expressed tumors in vivo. The limitations, such as the DAR range, aggregativity, solubility, and bystander effect of our conjugates, will also be explored" (lines 615–623).

Round 2
Reviewer 1 Report
Comments and Suggestions for Authors
In this revision, the authors have now decided that there is no immunologic aspect. Since immunosuppressed mice were used (line 527), this is a logical decision. The phrase ‘immuno’ appears only in a few of the references. It is claimed (line 32) that any photodynamic effects relating to this study are not significant. Efficacy of therapy is attributed solely to sonodynamic processes. There appear to be no studies carried out with light alone, so it is not clear how this is established. While the authors have abandoned the ‘immuno-’ concept for this report, they do refer (Ref. 41) to a prior publication relating to this approach, with ‘immuno-‘ in the title. Panel B of fig. 7 contains the ‘SIT’ abbreviation which should be removed.
Fig. 1 indicates that the dyes used are degraded by light or ultrasound. Light doses (60 J/sq cm) are high for in vitro studies. When PDT is used in vitro, effective light doses are usually in the 300-500 mJ/sq cm range. Animal studies require much higher light doses. Some photothermal effects were detected in vitro but it is difficult to predict what might happen in vivo. It is shown (Fig. 7)that sonodynamic therapy is can markedly inhibit tumor growth. Panel B indicates that only Groups 3 and 4 showed a significant anti-tumor effect. There appears to be no group that received PDT only. How is it concluded that there was no significant photodynamic effect?
It is noted (line 616) that treatment with ultrasound could be applied to the entire mouse, Fig. 6 does suggest that some conjugates localize only in sites of neoplasia, but it would be important to determine whether there was a sufficient concentration of any conjugate in host organs, e.g., liver, before whole animal ultrasound was attempted.
Author Response
Reviewer 1
Comment 1: In this revision, the authors have now decided that there is no immunologic aspect. Since immunosuppressed mice were used (line 527), this is a logical decision. The phrase ‘immuno’ appears only in a few of the references. It is claimed (line 32) that any photodynamic effects relating to this study are not significant. Efficacy of therapy is attributed solely to sonodynamic processes. There appear to be no studies carried out with light alone, so it is not clear how this is established. While the authors have abandoned the ‘immuno-’ concept for this report, they do refer (Ref. 41) to a prior publication relating to this approach, with ‘immuno-‘ in the title. Panel B of fig. 7 contains the ‘SIT’ abbreviation which should be removed.
Response 1: The study with light along (PDT) was carried out in our previous work [41]. Our conclusion, “We observe no significant contribution of PDT to the efficacy of the combined SDT and PDT, pointing out that SDT with 2ICy7–Ab is superior over PDT” is based on the comparison of the SDT results obtained in our current work with the PDT results in [41]. Now, to make our conclusions more evident, we added the tumor volumes for mice treated with light alone to Figure 7 (lines 331-341 in track mode file). These data were taken from [41]. Also, we apologize for the technical mistake, which is now corrected: “SIT” in Figure 7 is changed to “SDT”.
Comment 1: Fig. 1 indicates that the dyes used are degraded by light or ultrasound. Light doses (60 J/sq cm) are high for in vitro studies. When PDT is used in vitro, effective light doses are usually in the 300-500 mJ/sq cm range. Animal studies require much higher light doses. Some photothermal effects were detected in vitro but it is difficult to predict what might happen in vivo. It is shown (Fig. 7)that sonodynamic therapy is can markedly inhibit tumor growth. Panel B indicates that only Groups 3 and 4 showed a significant anti-tumor effect. There appears to be no group that received PDT only. How is it concluded that there was no significant photodynamic effect?
Response 2: We do not report on in vitro studies in our manuscript. The light dose of 63 J/cm2 is applied for animal studies (lines 267, 287, 294, 336-337, 358, 559). Upon the dye stability experiments (test-tube experiments without biological matter) represented in Figure 1 (lines 100-106), the light dose varied between zero and 120 J/cm2. It can be seen that the half-lives of the dyes were between 5-20 J/cm2. With light doses of 300-500 mJ/cm2 we cannot measure photostability because it requires too much time.
We agree that it is difficult to predict what might happen in vivo (including photothermal effects) in terms of photostability, and therefore such predictions were not carried out in our work.
The group that received PDT only is added to Figure 7 (lines 331-341). Notably, the light irradiation in mice experiments in the current manuscript was carried out in the same conditions (730 nm LED, 70 mW/cm2, 15 min, 63 J/cm2) as in [41]. Now, it is clearly seen that PDT only has a much lower cytotoxic effect than SDT alone. Therefore, we concluded that SDT with our conjugate is superior to PDT.
Comment 3: It is noted (line 616) that treatment with ultrasound could be applied to the entire mouse, Fig. 6 does suggest that some conjugates localize only in sites of neoplasia, but it would be important to determine whether there was a sufficient concentration of any conjugate in host organs, e.g., liver, before whole animal ultrasound was attempted.
Response 3: We agree with this suggestion and added the corresponding clarification to the Conclusions (lines 603-605-594).
Reviewer 2 Report
Comments and Suggestions for Authors
Accept in present form
Comments on the Quality of English LanguageMinor editing of English language required.
Author Response
Comment 1: Minor editing of English language required.
Response 1: The English language was verified by the English editor.
Round 3
Reviewer 1 Report
Comments and Suggestions for Authors
References to immunology have been removed. There appear to be two sets of images for Fig. 7. It seems likely that the bottom set represents the corrected version. Adding PDT to SDT does promote efficacy, but there are no data for PDT alone. It appears from Fig. 6 that the agents used concentrate at sites of neoplasia after 24 hours which would spare host organs from photo- or sonotoxicity. The limitation is that it was necessary to create an agent coupled to an antibody that aided in localization. Is it considered likely that an antibody-guided agent could be prepared for treatment of any spontaneously-arising tumor? These tend to be very heterogeneous (See Cancer Research 38, 3758-63, 1978).
With sonodynamic therapy, where penetration of ultrasound waves is difficult to direct, it will be important to establish that host organs, e.g., liver, are not ‘sensitized’. It may be feasible to create an agent that can be directed against BT-474 in a nude mouse. Whether similar effects could be produced in spontaneously-arising breast tumors is an unknown. Use of antibodies might be successful in sparing liver and other host organs from photo/sonotoxic effects, but how would spontaneously-arising tumors be treated? It is not clear whether anything reported here would be relevant to therapeutic applications in ‘the real world’. This should be discussed. The critical element is contained in the last sentence: ‘conjugated with corresponding antibodies . . ‘.. Corresponding to what?
Author Response
[IJMS] Manuscript ID: ijms-3195868
Title: Sonodynamic Therapy for HER2+ Breast Cancer with Iodinated Heptamethine Cyanine–Trastuzumab Conjugate
International Journal of Molecular Sciences
Dear Mr. Waylon Zhong,
We are grateful for reviewing our manuscript (Round 3) and appreciate reviewers’ valuable comments and suggestions. All of them have been addressed point-by-point in the revised version of our manuscript, which we provide with the track mode corrections and in the clean form.
Please find below our responses to comments and questions from reviewers.
We hope that now you will find this manuscript suitable for publication in International Journal of Molecular Sciences.
Sincerely,
Leonid Patsenker
Reviewer 1
Comment 1: References to immunology have been removed. There appear to be two sets of images for Fig. 7. It seems likely that the bottom set represents the corrected version.
Response 1: Yes, in the track mode version, the bottom set represents the corrected version, while the upper one is removed. The clean version shows only the right Fig. 7.
Comment 2: Adding PDT to SDT does promote efficacy, but there are no data for PDT alone.
Response 2: We kindly refer this reviewer to Fig. 7, where the efficacy results of PDT alone are shown as a violet curve designated as “and 2ICy7–Ab (PDT) [41]” and the citation on our previous work [41] is provided. It is clearly seen that the efficacy of PDT alone is minor compared to SDT alone.
Comment 3: It appears from Fig. 6 that the agents used concentrate at sites of neoplasia after 24 hours which would spare host organs from photo- or sonotoxicity. The limitation is that it was necessary to create an agent coupled to an antibody that aided in localization. Is it considered likely that an antibody-guided agent could be prepared for treatment of any spontaneously-arising tumor? These tend to be very heterogeneous (See Cancer Research 38, 3758-63, 1978). With sonodynamic therapy, where penetration of ultrasound waves is difficult to direct, it will be important to establish that host organs, e.g., liver, are not ‘sensitized’. It may be feasible to create an agent that can be directed against BT-474 in a nude mouse. Whether similar effects could be produced in spontaneously-arising breast tumors is an unknown. Use of antibodies might be successful in sparing liver and other host organs from photo/sonotoxic effects, but how would spontaneously-arising tumors be treated? It is not clear whether anything reported here would be relevant to therapeutic applications in ‘the real world’. This should be discussed.
Response 3: On our mind, it is hypothetically possible to develop antibody-guided agents for treatment of spontaneously-arising tumors, but there are some challenges to be solved, in particular heterogeneity (it is problematic to target all tumor cells with a single antibody), the development of tumor-specific antigens, and off-target effects (safety). Overcoming these problems goes beyond the work presented. It is important to note that SDT, as well as PDT, is a clinically approved methodology. A novel modification, which we provide in our work, is the employment of an iodinated cyanine as the sensitizer and equipment of this sonosensitizer with a target-specific antibody, which reduces side effects. We are aware of the spontaneously-arising heterogeneous tumors. Therefore, we added the corresponding comment to the conclusion, which addresses this issue, and provided the suggested citation (lines 598-600).
Comment 4: The critical element is contained in the last sentence: ‘conjugated with corresponding antibodies . . ‘.. Corresponding to what?
Response 4: To avoid misunderstandings, this statement is removed.